# Play-Friendly Communities in Nova Scotia, Canada: A Content Analysis of Physical Activity and Active Transportation Strategies

**DOI:** 10.3390/ijerph19052984

**Published:** 2022-03-03

**Authors:** Hilary A. T. Caldwell, Joshua Yusuf, Mike Arthur, Camille L. Hancock Friesen, Sara F. L. Kirk

**Affiliations:** 1Healthy Populations Institute, Dalhousie University, 1318 Robie Street, Halifax, NS B3H 3E2, Canada; hilary.caldwell@dal.ca (H.A.T.C.); js667453@dal.ca (J.Y.); mikearthur561@gmail.com (M.A.); chancockfriesen@childrensomaha.org (C.L.H.F.); 2School of Health and Human Performance, Dalhousie University, P.O. Box 15000, Halifax, NS B3H 4R2, Canada; 3Division of Pediatric Cardiothoracic Surgery, Children’s Health and Medical Centre, University of Nebraska Medical Center, 8200 Dodge Street, Omaha, NE 68114, USA

**Keywords:** municipality, policy, child-friendly, movement, active transportation, physical activity

## Abstract

The Play-Friendly Cities framework describes key municipal actions and indicators which support a community’s playability and can positively influence children’s health behaviors and quality of life. The purpose of this study was to conduct a content analysis of Nova Scotia physical activity (PA) and active transportation (AT) strategies by applying the *playability* criteria in the Play-Friendly Cities framework. Methods: PA and AT strategies from communities across Nova Scotia were assessed using the Play-Friendly Cities framework. Strategy content was analyzed based on indicators across four themes: participation of children in decision making, safe and active routes around the community, safe and accessible informal play environments, and evidence-informed design of formal play spaces. Results: Forty-two (28 PA,14 AT) strategies were reviewed and all included statements reflective of at least one indicator (8 ± 4; range: 1–14). Content about safe and active routes around the community was most prevalent (41 plans, 812 mentions), while participation of children in decision making was least frequently presented (18 plans, 39 mentions). Content about safe and accessible informal play environments (31 plans, 119 mentions) and evidence-informed design of formal play spaces (28 plans, 199 mentions) was also present. Conclusions: All PA and AT strategies included some content reflective of a Play-Friendly City; however, there was great variability in the number of included indicators. This summary provides key information on opportunities, such as increasing meaningful involvement of children in decision making, that can inform future municipal actions and policies to improve a community’s playability.

## 1. Introduction

Physical activity (PA) participation in childhood is positively associated with favorable measures of physical, mental and psychosocial health [1]. There is also growing evidence that playing outdoors or using active transportation (AT) are associated with increased PA in children and youth [2,3]. Despite the benefits associated with PA, AT, and outdoor play, only 39% of Canadian children and youth engage in enough PA daily to achieve health benefits [4]. Children and youth benefit from living in communities that promote outdoor play and walking in their design, programs, and policies [5]. For example, neighborhood design, availability of recreation facilities, and sidewalks are positively associated with increased outdoor play in children and youth [6]. Given the importance of PA to health and well-being, it is recommended that governments at all levels create new, or expand existing, efforts to promote active living. This can be done through creating and maintaining playgrounds, parks and green spaces, and funding active school travel programs [5]. Innovative strategies, programs, and policies are needed to ensure children and youth have access to convenient, fun, and safe PA opportunities, supported by a safe, accessible built environment. Furthermore, measuring the impact of such actions is essential to understand how municipal government policies and investments are working to create play-friendly spaces for young people to grow and develop.

Around the world, municipalities are implementing play-friendly designs [7]. Let’s Get Moving, Canada’s Common Vision for increasing physical activity and reducing sedentary behavior, highlights the need to open up play, by increasing opportunities for self-directed, safe play in outdoor settings. For example, the Vision recommends that municipalities re-examine laws that prohibit safe, outdoor play (e.g., tobogganing, street hockey); that schools develop shared use agreements to increase community access to school sport facilities; and that communities host open street events to encourage more walking and cycling [8]. In 2019, the Canadian Public Health Association released the Play-Friendly Cities Framework of Action, which describes the characteristics of a community that supports unstructured play in its design and policies [9]. This framework expands upon UNICEF’s Child-Friendly City framework that describes a community that supports the voices, needs, and priorities of children in its public policies, programs, and decisions [10]. Play-Friendly Cities consider children’s well-being and access to play in their design and are developed to promote the involvement and active participation of children and youth in decision making. In addition, they recognize the role of infrastructure design, such as parks, playgrounds, and opportunities for AT, in child development and the necessity for children to access such infrastructure in a healthy and safe way. Lastly, a Play-Friendly City prioritizes opportunities for PA and play in everyday spaces, not only dedicated play spaces, to meet the diverse needs of all children [9]. The Play-Friendly Cities Framework of Action is comprised of 20 recommended actions (indicators) across four playability criteria (themes): (1) Participation of children in decision-making; (2) Safe and active routes around the community; (3) Safe and accessible informal play environments, and (4) Evidence-informed design of formal play spaces (see Table 1). 

The CPHA Play-Friendly Cities framework is a promising, evidence-informed framework to assess how municipalities are planning or implementing its recommended actions. To our knowledge, this framework has not yet been applied to the analysis of municipal PA or AT strategies or plans in Canada. A previous study in Saskatchewan, Canada reviewed official community plans or policies supportive of PA in 17 cities and found that the most common policies were related to residential neighborhood plans, downtown cycling and pedestrian plans, plans to improve active recreation, joint-use agreement between community and schools, school travel plans, and AT plans [11]. Except for those associated with school travel plans and AT, the indicators included in the Saskatchewan official community plans were not reflective of the actions included in the Play-Friendly Cities framework. Internationally, a study from Sweden reported on municipal PA promotion policies and determined that the most frequently reported measures were related to increasing AT to school, winter bike path maintenance, separation of bike lanes from traffic and the inclusion of PA objectives in municipal school plans [12]. The researchers did not investigate whether municipal plans included physical environment features related to parks, playgrounds, or other spaces that support PA. Therefore, there is a gap in our understanding of how to evaluate play-friendly actions within Canadian municipalities, as well as in how this new framework might be applied in practice. 

Given the benefits of PA, AT, and outdoor play participation for children and youth, and the influence of the built environment on PA opportunities, it is important to better understand how communities are supporting PA and active play. Let’s Get Moving Nova Scotia, a plan to encourage Nova Scotians to include more movement in their daily lives, highlights community plans as an important strategy to increase PA levels. The purpose of this study was to assess actions laid out in municipal PA and AT strategies in Nova Scotia to determine how well they aligned with the indicators in the Play-Friendly Cities Framework of Action. We conducted a content analysis of available plans to identify the utility of the framework and opportunities to strengthen actions at the municipal level within the Canadian context. 

## 2. Materials and Methods

### 2.1. Study Area

Nova Scotia is a province on the east coast of Canada with a total size of 52,942 km^2^ and a population of 923,598 [13]. Nova Scotia is made up of 50 municipalities (four regional municipalities, 26 towns, 20 county or district municipalities), and 13 Mi’kmaq First Nation communities [13,14]. Municipalities across Nova Scotia that have developed community plans to promote physical activity represent the unit of analysis for this study.

### 2.2. Data Sources 

Available PA (*n* = 28) and AT (*n* = 14) strategies of municipalities and communities in Nova Scotia were included in this study. Municipal websites were searched for copies of strategies. If strategies were not publicly available on websites, municipal PA leaders or recreation staff were contacted, and a copy of the strategy was requested. Some communities did not have physical activity strategies but had publicly available recreation plans, which were included if a PA strategy was not available for that municipality.

### 2.3. Data Extraction and Analysis

The municipal strategies were analyzed with content analysis, a qualitative method to determine the presence of certain concepts within some given text [15]. Content analysis of plans was completed independently by two reviewers (HATC and JY) in NVivo Version 12Pro to confirm relevance and categorization according to the Play-Friendly Cities recommended actions. 

Reviewers met regularly to review coding of plans and determine consensus among discrepancies in coding plans. Once consensus was reached across all plans, the reviewers extracted the data for each action/indicator. For each plan, the name of the community, type of community (regional, district or county municipality, town, or First Nation), community population, and year the plan was developed were extracted and summarized. The number of indicators included in each individual plan was determined. For each indicator, the number of strategies that included that indicator and the total number of actions that reflected that indicator across all strategies were determined. 

During the review process, we identified categories to capture actions related to the Play-Friendly Cities framework that were not explicitly reflective of the recommended actions. As a result, an ‘Other’ category for each theme was added. We also added a code titled ‘Youth engagement or consultation in plan development’ as many plans included youth voice in collecting background information for the plans, but not as an action in their strategies and these statements could therefore not be coded as one of the recommended actions in the ‘Participation of children in decision-making’ theme. Lastly, we added an indicator for ‘equity, diversity, and inclusion’ to reflect the values and priorities of Nova Scotia’s Let’s Get Moving plan about removing barriers that keep people from participating in PA and considering the needs of the less active or other priority groups [16].

## 3. Results

The community, community types, populations, year, and number of indicators for each strategy are included in Table 2. All strategies were completed from 2008 to 2021 (year of one strategy was unknown). The average number of indicators per PA strategy was 9 ± 3 (range: 3 to 14) and the average number per AT strategy was 5 ± 2 (range: 1 to 8) indicators. 

Table 3 includes a complete list of indicators, the number of strategies that included each indicator, and the frequency that each indicator was mentioned across all strategies and individually for PA and AT strategies. The most frequently mentioned indicator across all plans was ‘safe and active routes around the community other’ (451 mentions), while ‘equity, diversity, and inclusion’ was most frequently included in PA strategies (285 mentions, 31 strategies) (Figure 1) and ‘design streets to safely accommodate all users (pedestrians, cyclists, transit, vehicles)’ was most frequently included in AT strategies (268 mentions, 13 strategies) (Figure 2). The least frequently mentioned indicators were those included in the ‘participation of children in decision making’ theme (Figure 1) and ‘youth engagement or consultation in plan development’ was the most frequently mentioned indicator within this theme (*n* = 24).

### 3.1. Participation of Children in Decision Making 

Actions related to this theme were infrequently reported. Fourteen PA and five AT strategies included youth input in developing their plans, but very few included actions and objectives that specifically engaged youth in decision making. For example, Antigonish’s PA strategy included “a series a focus groups and informal surveys with Antigonish children and youth indicated their top choices for physically active pursuits were…” and Argyle’s AT Plan mentioned “two youth workshops with student councils”. The ‘develop consultation processes to include children in municipal decision making’ action was included in 10 PA strategies. For example, Acadia First Nation mentioned, “connect Elders and youth to create opportunities to share traditional knowledge and skills” and Berwick included “create a Youth Advisory group to meet throughout the year to brainstorm ideas and design programs/events”. 

### 3.2. Safe and Active Routes around the Community

Indicators in this domain were frequently mentioned in both PA and AT strategies. Fifteen PA strategies included ‘Provide active commute programming to and from school, including with reduced supervision’ mentions, such as “assess the schools in the area with the Safe Routes to School Travel Planning Guide” (Barrington), “continue with walking school bus” (Berwick), and “work with schools to incorporate AT into curricula” (Kentville). In the AT strategies, 13 of the 14 plans included actions to ‘Design streets to safety accommodate all users’, such as: “the AT network should provide a safe environment for all users through well-maintained infrastructure. This infrastructure must be considerate of wheelchairs, strollers and slow-moving pedestrians” (Bridgewater) and “the municipality should consider protected bicycle lanes wherever there are candidate bicycle routes…and aim to implement at least one protected bicycle lane pilot project in the next five years” (Halifax). Interestingly, no actions for ‘implement measures to reduce (parental) vehicular traffic in school zones’ were included in the AT strategies. Both PA and AT strategies included a high number of actions in the ‘other’ category. Examples from AT plans include: “share the road public education campaign” (Annapolis), “place bicycle racks at important civic locations and businesses” (Inverness), and “Install share the route signage…” (Yarmouth). PA strategies included the following examples: “support the development of a signage/way finding system to mark routes and distances at trailheads and along trails” (Antigonish Communities in Movement), “maintain the trails” (Barrington), “expand community bike program” (Cumberland), “address safety concerns and other barriers to AT within the community” (Glooscap), and “sidewalk snow clearing” (Pictou). 

### 3.3. Safe and Accessible Informal Play Environments

‘Safe and accessible informal play environments’ indicators were infrequently included in AT strategies and these mentions focused on the intersection of play spaces and AT routes, such as “use the Harvest Moon Trailway as a platform for play” (Kentville) as an ‘update everyday public space to be inclusive of child play’ action. The indicator ‘Provide programming to encourage safe play on streets without motor vehicle traffic’ was not included in any AT or PA strategies. ‘Preserve play-friendly outdoor green space’ actions were mentioned in 16 PA strategies and included actions related to community clean-up days, supporting recreational facilities as venues for year-round recreation, and maintenance of parks, playgrounds, and sports fields. Sixteen PA strategies included actions related to ‘update everyday public space to be inclusive of child play’, such as “increase greenspace and opportunities for spontaneous outdoor free play by the creation of play spaces throughout the community” (Berwick) and “explore usage of local facilities, such as community halls, in non-traditional ways to promote PA and connectivity” (Queens). Several actions for this domain were classified as ‘other’, including: “Increase support for the construction of an ‘All Wheels’ Park” (Antigonish), “Increase the number of unstructured outdoor play opportunities” (Argyle), “Create partnerships and be aware of available grants for the construction of new structures such as a natural playground or skate park” (Barrington), and “work with the school community to facilitate children and youth utilizing green spaces near schools” (Truro). 

### 3.4. Evidence-Informed Design of Formal Play Spaces

AT strategies did not include any mentions of the actions within this theme; however, all indicators were mentioned in PA strategies. The ‘limit unnecessary rules in play spaces in order to encourage thrilling and challenging play’ indicator was only mentioned in one plan: “encourage unstructured outdoor play during all seasons promoting a “risk tolerant” culture within the municipality” (Antigonish Communities in Movement). ‘Offer play spaces with loose parts, natural elements, and pop-up adventure activities’ were included in 24 PA strategies. Examples include: “Support after school free-play programs to strengthen participation and interest in unstructured play” (Antigonish), “Create a playground program that offers a mobile “Loose Parts” workshop” (Cumberland), “implement play boxes in strategic outdoor locations to encourage unstructured play for children and families” (Oxford), and “Identify and promote PA opportunities that do not require registration, advanced skills or other commitments (e.g., drop-in programs)” (Richmond). ‘Apply universal design principles to develop play spaces that are accessible for all abilities’ actions were included in 17 PA strategies and examples include: “create trails that are wheelchair/stroller accessible” (Barrington), “consult with designers and planners that specialize in accessibility” (Shelburne), and “support the development of a Boundless (inclusive) Playground” (Yarmouth). The indicator ‘Ensure play spaces offer age-appropriate challenges across many development stages’ was included in eight strategies and mentioned mostly in initiatives targeting older youth, such as skate parks; however, Truro’s PA strategy mentioned younger age groups too (“Assess and adapt/develop outdoor playgrounds to ensure they can be safely used by children under five years of age”). 

### 3.5. Equity, Diversity, and Inclusion 

Equity, diversity, and inclusion actions were included in 27 PA and four AT strategies. Strategies included specific actions for various groups who may face barriers to participation in PA and AT, including people with disabilities, women, girls, females, new Canadians, Mi’kmaq populations, seniors, low-income families, and teenagers. Numerous plans included actions related to reducing or eliminating barriers to participation in PA or recreation, such as reduced or free fees, equipment loan programs (e.g., bikes, kayaks), provision of adaptive equipment and technology, transportation services, provision of childcare, offering family programs, offering culturally relevant programs, and increasing the accessibility of facilities and community infrastructure (e.g., sidewalks, trails). 

## 4. Discussion

Creating the conditions and environments that support movement and play among children and youth is a prerequisite for a physically active population [5]. This study assessed the content of PA and AT strategies from communities across Nova Scotia to determine their alignment with the actions proposed in the Play-Friendly Cities Framework of Action [9]. To our knowledge, this is the first study to assess the playability of municipal PA and AT strategies based on this framework. All plans included some content reflective of the Play-Friendly Cities Framework; however, this ranged across plans and is likely because strategies focused on PA promotion for the whole population and not specifically children and youth. PA and AT strategies frequently mentioned actions related to the ‘Safe and active routes around the community’ theme and least frequently mentioned actions related to the ‘Participation of children in decision making’ theme. The results of this study highlight how many communities in Nova Scotia proposed actions related to playability, but also highlighted actions that can be incorporated in future PA and AT strategies. 

The PA plans we reviewed included a wide range of indicators across all themes in the Play-Friendly Cities Framework. Some of the included indicators aligned well with the Nova Scotia’s Let’s Get Moving strategy, particularly those related to school AT, community-school partnerships to support PA, community walking and cycling groups, accessibility of facilities and buildings, and addressing financial barriers to participation [16]. The Play-Friendly Cities Framework emphasizes the availability, accessibility, safety, and challenge of outdoor play spaces. Almost all PA strategies included the indicator ‘offer play spaces with loose parts, natural elements, and pop-up adventure activities’ by describing actions related to ‘try-it’ days or drop-in sessions for activities like snow shoeing, hiking, or kayaking. A study that reviewed PA-related policies in Saskatchewan community plans reported policies related to residential neighborhood plans, cycling and pedestrian plans, and joint-use agreements between communities and schools were most common [11]. Similarly, actions related to ‘Safe and active routes around the community’ were reported in all but one PA strategy. We observed numerous actions related to joint-use agreements between communities and schools. However, without a dedicated indicator in the framework, we categorized these actions in the ‘Evidence-informed design of formal play spaces other’ because research supports joint-use agreements as a positive strategy for PA promotion, particularly in rural areas where there may not be alternative sport or recreation facilities [17]. 

AT strategies primarily included actions related to the ‘design of streets to safely accommodate all users (pedestrians, cyclists, transit, vehicles)’ and the ‘other’ indicators. Most AT strategies included actions related to active commute programming at schools, such as implementing school travel planning or walking school buses. The Nova Scotia Let’s Get Moving Plan also includes goals to increase opportunities for students to commute to school by walking or cycling [16]. For the most part, actions in the plans related to the encouragement in, or facilitation of, active school travel, but not to the reduction of parental vehicular traffic at schools, a strategy that should be coupled with the former measures to promote active school travel [18]. 

Across both PA and AT strategies, mentions of actions related to the inclusion of children and youth were very minimal. In the plans that mentioned youth involvement in decision making, the actions related to youth advisory councils, connections between youth and Elders in Mi’kmaq communities, or youth leadership opportunities. As outlined in the Play-Friendly Cities Framework, children and youth are valuable community members who can make meaningful contributions to municipal decisions related to community design [10]. When youth were consulted about the play spaces in a community in British Columbia, they learned they have a right for their voices to be heard in their communities and they enjoyed sharing their ideas about road safety, playground hazards, and play spaces [19]. When engaging with children for decision-making, practitioners should consider using age-appropriate strategies that focus on children’s desires for play, creativity, movement, and exploration [20]. Future updates to PA and AT strategies should consider how youth can be further involved in municipal decision making about PA and AT. 

In this study, we added an additional indicator related to equity, diversity, and inclusion to align with Nova Scotia’s Let’s Get Moving Strategy goal to “enhance opportunities and address inclusion” [16]. Equity, diversity, and inclusion actions included programming for girls and women, equipment loan programs, free or reduced-cost activities, and the physical accessibility of play spaces. These actions are important to address the lower PA levels and steeper declines in PA participation among girls versus boys in childhood [21] and cost-related barriers to accessing sport and recreation facilities and obtaining equipment [22]. ‘Apply universal design principals to develop play spaces that are accessible for all abilities’ actions were mentioned in about half of PA strategies despite evidence that inaccessible spaces are a commonly cited barrier for children with disabilities wanting to participate in leisure-time PA [23,24]. Based on our findings, future municipal strategies should consider how actions can promote equity, diversity, and inclusion. 

A strength of this study is the application of the Play-Friendly Cities Framework of Action to assess the content of Nova Scotia PA and AT strategies using a rigorous, systematic process. In addition, we assessed actions related to equity, diversity, and inclusion and highlight this topic as an important consideration in future municipal strategies. There are a few limitations. At the time of our study, only two-thirds of communities had available strategies and therefore we cannot provide a province-wide evaluation of such strategies. In addition, we did not assess what, or how, actions in the strategies were currently being implemented. We identified that municipal events (e.g., fairs, festivals) and trails were not included in the Play-Friendly Cities Framework; however, these topics were frequently mentioned in the strategies, and we believe would align well with the Framework. Finally, municipalities did not use the playability criteria in the development of their strategies and therefore would not necessarily have been familiar with the concepts. Our use of the framework to conduct this analysis is therefore post-hoc but may be of benefit when strategies are updated.

## 5. Conclusions

In conclusion, we determined that all available Nova Scotia PA and AT strategies included some actions related to the Play-Friendly Cities Framework, but their presence varied greatly across strategies. Upon review of the actions included in the strategies, we added an indicator related to equity, diversity, and inclusion as most strategies included important actions related to this indicator. We recommend future iterations of the CPHA framework also add indicators related to this topic and the focus be expanded to represent communities more broadly rather than cities specifically. Our analysis provides a baseline of the content of Nova Scotia PA and AT strategies, and future work can explore if, or how, actions are being implemented in communities. We anticipate that the results of the study can be used by municipal units that are working on similar actions and to educate municipal officials on gaps such as play streets, youth engagement, and reducing parental vehicle traffic in school zones. 

## Figures and Tables

**Figure 1 ijerph-19-02984-f001:**
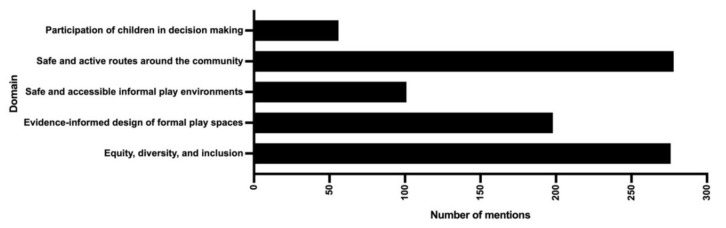
Total number of mentions of indicators in each Play-Friendly Cities domain for Nova Scotia Physical Activity strategies.

**Figure 2 ijerph-19-02984-f002:**
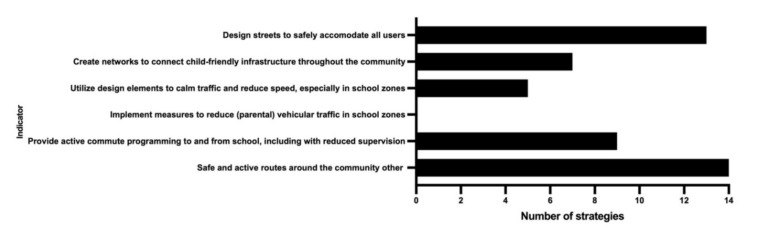
Number of strategies that include each indicator in the Play-Friendly Cities Safe and Active Routes Around the Community domain for Nova Scotia active transportation strategies.

**Table 1 ijerph-19-02984-t001:** Framework of Action for a Play-Friendly City.

Playability Criteria (Themes)	Description	Number of Recommended Actions/Indicators
Participation of children in decision-making	Municipalities actively engage diverse groups of children in relevant political matters	5
Safe and active routes around the community	Municipalities provide safe, accessibly infrastructure that facilitates children’s active and/or independent travel around their neighborhood, especially to and from school	5
Safe and accessible informal play environments	Municipalities design public space to promote unstructured play in children’s everyday natural environments	4
Evidence-informed design of formal play spaces	Municipalities use best practice of play space design to provide challenging opportunities for children’s unstructured play	6

Note: Adapted from the Canadian Public Health Association, 2019.

**Table 2 ijerph-19-02984-t002:** Communities, year of strategy, population, and frequency and prevalence of indicators included per physical activity or active transportation strategy.

	Community	Type of Community	Year of Strategy	Population (2016)	Indicators, *n* (%)
Physical Activity Strategies (*n* = 28)	Acadia	First Nation	2016	1664	8 (30.8%)
	Amherst	Town	2018	9415	9 (34.6%)
	Antigonish	County Municipality	2019	14585	14 (53.8%)
	Antigonish	Town	2017	4365	11 (42.3%)
	Argyle	District Municipality	2016	7900	9 (34.6%)
	Barrington	District Municipality	2012	6645	12 (46.2%)
	Berwick	Town	2017	2510	14 (53.8%)
	Cumberland	County municipality	2018	19405	11 (42.3%)
	East Hants	District Municipality	2019	22455	3 (11.5%)
	Glooscap	First Nation	2019	393	5 (19.2%)
	Kentville	Town	2018	6270	11 (42.3%)
	Lunenburg	District Municipality	2014	24860	11 (42.3%)
	Membertou	First Nation	2015	1573	9 (34.6%)
	Middleton	Town	2013	1830	4 (15.4%)
	New Glasgow	Town	2019	9075	4 (15.4%)
	Oxford	Town	2019	1190	10 (38.5%)
	Pictou	County Municipality	2019	20690	7 (26.9%)
	Potlotek	First Nation	2019	773	7 (26.9%)
	Queen’s	Regional Municipality	2020	10305	7 (26.9%)
	Richmond	County Municipality	2009	8460	14 (53.8%)
	Shelburne-Lockeport	District Municipality (Shelburne) and Towns (Shelburne and Lockeport)	2020	6560	7 (26.9%)
	Shelburne	District Municipality (Shelburne)	2019	4290	10 (38.5%)
	Truro	Town	2021	12260	8 (30.8%)
	Victoria	County Municipality	2017	6555	14 (53.8%)
	We’koqma’q	First Nation	2018	1031	8 (30.8%)
	West Hants	Regional Municipality	2018	15370	11 (42.3%)
	Wolfville	Town	2013	4195	10 (38.5%)
	Yarmouth	District Municipality & Town	2014	16005	11 (42.3%)
Active Transportation Strategies (*n* = 14)	Amherst	Town	2018	9415	3 (11.5%)
	Annapolis	County Municipality	Unknown	18255	2 (7.7%)
	Argyle	District Municipality	2020	7900	6 (23.1%)
	Avon Region (West Hants, Windsor, Hantsport)	Regional Municipality (West Hants), Towns (Windsor and Hantsport)	2015	16005	7 (26.9%)
	Bridgewater	Town	2008	8535	2 (7.7%)
	Cape Breton	Regional Municipality	2013	94285	1 (3.8%)
	Halifax	Regional Municipality	2014	403130	4 (15.4%)
	Inverness	County Municipality	2018	13190	5 (19.2%)
	Kentville	Town	2019	6270	8 (30.8%)
	Lunenburg	Town	2013	2260	8 (30.8%)
	Port Hawkesbury	Town	2014	3215	5 (19.2%)
	Sherbrooke/St. Mary’s	District Municipality (St. Mary’s)	2016	2233	5 (19.2%)
	Stellarton	Town	2019	4210	4 (15.4%)
	Yarmouth	District municipality and Town	2010	16005	6 (23.1%)

Note: Populations of regional municipalities, towns, county municipalities, and district municipalities based on 2016 Census data and populations of Mi’kmaq First Nation communities based on Indigenous and Northern Affairs data.

**Table 3 ijerph-19-02984-t003:** Number of mentions and strategies that include each Play-Friendly Cities indicator in Nova Scotia physical activity and active transportation strategies.

Playability Criteria (Themes)	Indicator		All Strategies (*n* = 42)	PA Strategies (*n* = 28)	AT Strategies (*n* = 14)
Participation of children in decision making	Develop consultation processes to include children in municipal decision making	Strategies, *n*	10	10	0
		Mentions, *n*	18	18	0
	Involve children in political matters that affect them in meaningful ways	Strategies, *n*	0	0	0
		Mentions, *n*	0	0	0
	Include the voices of children of diverse ages, abilities, and perspectives	Strategies, *n*	1	1	0
		Mentions, *n*	1	1	0
	Utilize multiple mechanisms and formats for children to voice their perspective	Strategies, *n*	1	1	0
		Mentions, *n*	1	1	0
	Provide direct access to decision-makers through municipal roles for children	Strategies, *n*	3	3	0
		Mentions, *n*	3	3	0
	Participation of children in decision making other	Strategies, *n*	12	11	1
		Mentions, *n*	16	15	1
	Youth engagement or consultation in plan development	Strategies, *n*	19	14	5
		Mentions, *n*	24	18	6
Safe and active routes around the community	Design streets to safely accommodate all users (pedestrians, cyclists, transit, vehicles)	Strategies, *n*	24	11	13
		Mentions, *n*	268	24	244
	Create networks to connect child-friendly infrastructure throughout the community	Strategies, *n*	13	6	7
		Mentions, *n*	38	11	27
	Utilize design elements to calm traffic and reduce speed, especially in school zones	Strategies, *n*	7	2	5
		Mentions, *n*	16	2	14
	Implement measures to reduce (parental) vehicular traffic in school zones	Strategies, *n*	1	1	0
		Mentions, *n*	1	1	0
	Provide active commute programming to and from school, including with reduced supervision	Strategies, *n*	24	15	9
		Mentions, *n*	38	23	15
	Safe and active routes around the community other	Strategies, *n*	41	27	14
		Mentions, *n*	451	217	234
Safe and accessible informal play environments	Preserve play-friendly outdoor green space	Strategies, *n*	18	16	2
		Mentions, *n*	38	35	3
	Provide programming to encourage safe play on streets without motor vehicle traffic	Strategies, *n*	0	0	0
		Mentions, *n*	0	0	0
	Remove municipal by-laws that discourage or prohibit street play	Strategies, *n*	2	1	1
		Mentions, *n*	2	1	1
	Update everyday public space to be inclusive of child play	Strategies, *n*	18	16	2
		Mentions, *n*	26	21	5
	Safe and Accessible Informal Play Environments Other	Strategies, *n*	22	20	2
		Mentions, *n*	53	44	9
Evidence-informed design of formal play spaces	Offer play spaces with loose parts, natural elements, and pop-up adventure activities	Strategies, *n*	24	24	0
		Mentions, *n*	87	87	0
	Ensure play spaces offer age-appropriate challenges across many development stages	Strategies, *n*	8	8	0
		Mentions, *n*	9	9	0
	Apply universal design principles to develop play spaces that are accessible for all abilities	Strategies, *n*	17	17	0
		Mentions, *n*	43	43	0
	Adapt play spaces to endure and be safe in typical weather conditions	Strategies, *n*	8	7	1
		Mentions, *n*	9	8	1
	Limit unnecessary rules in play spaces in order to encourage thrilling and challenging play	Strategies, *n*	1	1	0
		Mentions, *n*	1	1	0
	Develop play spaces through community input from diverse children and parents/caregivers	Strategies, *n*	2	2	0
		Mentions, *n*	2	2	0
	Evidence-Informed Design of Formal Play Spaces Other	Strategies, *n*	18	18	0
		Mentions, *n*	48	48	0
Equity, diversity, and inclusion		Strategies, *n*	31	27	4
		Mentions, *n*	285	276	9

PA: physical activity; AT: active transportation.

## Data Availability

The dataset supporting the conclusion of this article is available from the authors upon reasonable request and the completion of a data transfer agreement.

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
