# Peer review of "Play-Friendly Communities in Nova Scotia, Canada: A Content Analysis of Physical Activity and Active Transportation Strategies"

_ijerph, 2022, doi:10.3390/ijerph19052984_

Round 1

Reviewer 1 Report

The authors examine the physical activity (PA) and active transportation (AT) strategies in Nova Scotia, Canada. Based on the playability criteria of the Play Friendly Cities Framework set by the Canadian Public Health Association (CPHA) in 2019, the authors qualitatively evaluate municipalities’ and communities’ strategies for PA and AT in Nova Scotia with content analysis. I think it is important to know how municipal governments and communities are up to the principles to promote outdoor activities among children and youths. However, I believe the authors should strengthen the manuscript in many ways before publication.

First of all, the authors should have elaborated the overall research question more clearly. In a nutshell, the paper explores local PA and AT strategies in Nova Scotia against the criteria set by the 2019 CPHA. However, there is no explanation of why the authors anchor their analysis around the 2019 Play Friendly Cities Framework of Action. Is this the only criteria to evaluate the local strategies? What can we expect to accomplish by evaluating PA and AT strategies with the framework? Without providing any explanations about these somewhat foundational study setups, the authors directly jump into the study to evaluate the local strategies.

Second, as the analysis goes, the four playability criteria of the Play Friendly Framework were further classified into 20 indicators, with which the authors scrutinize the local PA and AT strategies. All of a sudden, however, the authors added the “equity, diversity, and inclusion” indicator simply because they encounter these comments frequently across strategies. Despite the importance of the indicator, I think this is the weakest point of the paper. It basically debates the overall study plan to include this indicator. In a way, this point is related to the first criticism that the authors should work more on the study design. If the Play Friendly Framework provides good criteria to evaluate local strategies, why does it not include such an important indicator in the first place? Or, is there any way to classify the indicator into one of the existing themes of the framework?

Third, the authors should also explain why we need to focus on the strategies, not the de facto PA and AT. It is possible that the actual PA and TA status is way behind the strategies. It is also theoretically possible that the de facto PA and TA would be better than the plan. The issue is not about the focus on strategies but about the lack of explanation on why we explore strategies over the de facto PA and TA.

Lastly, tables are not that reader-friendly. A dotted line between themes would be helpful.

The efforts to collect and analyze the data are highly valuable. I hope that the authors find my comments helpful in strengthening the manuscript. Thanks for reading my comment.

Author Response

Reviewer 1 comment: The authors examine the physical activity (PA) and active transportation (AT) strategies in Nova Scotia, Canada. Based on the playability criteria of the Play Friendly Cities Framework set by the Canadian Public Health Association (CPHA) in 2019, the authors qualitatively evaluate municipalities’ and communities’ strategies for PA and AT in Nova Scotia with content analysis. I think it is important to know how municipal governments and communities are up to the principles to promote outdoor activities among children and youths. However, I believe the authors should strengthen the manuscript in many ways before publication.

First of all, the authors should have elaborated the overall research question more clearly. In a nutshell, the paper explores local PA and AT strategies in Nova Scotia against the criteria set by the 2019 CPHA. However, there is no explanation of why the authors anchor their analysis around the 2019 Play Friendly Cities Framework of Action. Is this the only criteria to evaluate the local strategies? What can we expect to accomplish by evaluating PA and AT strategies with the framework? Without providing any explanations about these somewhat foundational study setups, the authors directly jump into the study to evaluate the local strategies.

Our response: Thank you for this comment. We have now added further information throughout the introduction about the purpose of the study and our rationale for using the Play Friendly Cities framework (lines 63-70, 109-111, 121-123).

Reviewer 1 comment: Second, as the analysis goes, the four playability criteria of the Play Friendly Framework were further classified into 20 indicators, with which the authors scrutinize the local PA and AT strategies. All of a sudden, however, the authors added the “equity, diversity, and inclusion” indicator simply because they encounter these comments frequently across strategies. Despite the importance of the indicator, I think this is the weakest point of the paper. It basically debates the overall study plan to include this indicator. In a way, this point is related to the first criticism that the authors should work more on the study design. If the Play Friendly Framework provides good criteria to evaluate local strategies, why does it not include such an important indicator in the first place? Or, is there any way to classify the indicator into one of the existing themes of the framework?

Our response: Thank you for this comment. We have further justified our use of the Play Friendly Cities Framework in the introduction based on your previous comment. We believe the inclusion of the ‘equity, diversity, and inclusion’ indicator was necessary for several reasons. First of all, it is well documented that physical activity levels are lower in numerous equity-deserving groups (i.e., girls, women, persons with disabilities, new Canadians) and that these groups face additional barriers when accessing physical activity opportunities. Secondly, as described in our paper on lines 169-173, it is a priority of Nova Scotia’s Let’s Get Moving strategy to consider the needs of those less active and to remove barriers to participation. We did not feel this indicator was adequately captured in the Play Friendly Cities Framework and recommend that a similar indicator be added to further iterations of the Framework (lines 417-420). If the reviewer feels strongly that this be removed, we can remove it; however, we believe it is important to highlight and share the actions that are reflective of this indicator so that they can be considered for inclusion in future work.

Reviewer 1 comment: Third, the authors should also explain why we need to focus on the strategies, not the de facto PA and AT. It is possible that the actual PA and TA status is way behind the strategies. It is also theoretically possible that the de facto PA and TA would be better than the plan. The issue is not about the focus on strategies but about the lack of explanation on why we explore strategies over the de facto PA and TA.

Our response: Thank you for this comment. We have clarified why our approach is needed and added to the literature on this topic.

Reviewer 1 comment: Lastly, tables are not that reader-friendly. A dotted line between themes would be helpful.

Our response: Thank you for this suggestion. We have updated the tables to reflect the templates provided by the journal.

Reviewer 1 comment: The efforts to collect and analyze the data are highly valuable. I hope that the authors find my comments helpful in strengthening the manuscript. Thanks for reading my comment.

Our response: Thank you for reviewing our manuscript. We hope we have addressed the comments appropriately and believe the changes have strengthened the manuscript.

Reviewer 2 Report

Title: “Play-friendly communities in Nova Scotia, Canada: a content 1 analysis of physical activity and active transportation strategies”.

I would thank the Editor for the possibility to review this manuscript. In general, I think that the manuscript is well-written and clear. The background is exhaustive and supported by relevant references. I did not recommend English revision. Prior to the publication, I recommend a few corrections. 

The introduction section is well-written, exhaustive, and supported by relevant references. I recommend shortening Active transportation with AT at line 45.

Methods: I suggest changing in italics the sentence “Data Extraction and Analysis” at line 118.

Results: In this section, it would be better to delete the hashtag # before all the “strategies” and “mention” in the table. 

line 166: I suggest removing the bold type. 

Discussion: I suggest changing the -ing form in this section (i.e., line 299 encouraging or facilitating, line 299 reducing, line 310 sharing, etc…) with a better form, such as a passive form.

Conclusion: This paragraph is well-written. I did not recommend any revision.

Author Response

Reviewer 2 comment: I would thank the Editor for the possibility to review this manuscript. In general, I think that the manuscript is well-written and clear. The background is exhaustive and supported by relevant references. I did not recommend English revision. Prior to the publication, I recommend a few corrections. 

The introduction section is well-written, exhaustive, and supported by relevant references. I recommend shortening Active transportation with AT at line 45.

Our response: Thank you for this feedback. We have updated Active Transportation to be AT online 45.

Reviewer 2 comment: Methods: I suggest changing in italics the sentence “Data Extraction and Analysis” at line 118.

Our response: Thank you, this has been italicized.

Reviewer 2 comment: Results: In this section, it would be better to delete the hashtag # before all the “strategies” and “mention” in the table. 

Our response: The tables have been updated to include ‘n’ to represent the number of strategies. e

Reviewer 2 comment: line 166: I suggest removing the bold type. 

Our response: Thank you, the bold font has now been removed.

Reviewer 2 comment: Discussion: I suggest changing the -ing form in this section (i.e., line 299 encouraging or facilitating, line 299 reducing, line 310 sharing, etc…) with a better form, such as a passive form.

Our response: Thank you for this feedback. We have updated the form of several words throughout the discussion.

Reviewer 2 comment: Conclusion: This paragraph is well-written. I did not recommend any revision.

Our response: Thank you for the positive feedback on our conclusion paragraph.

Reviewer 3 Report

First, I would like to thank you for trusting me to review your manuscript. It is essential to determine the existing strategies for the promotion of physical activity carried out by institutions and to estimate whether these are adequate for that purpose or can be improved. In any case, there are some questions that should be considered prior to publication. 

  1. Introduction
    Line 47: the first time the authors mention AT they should explain that it is an active transport.
    Line 87: why do the authors mention a study conducted in Sweden? The authors should justify the connection to the study area. 
  2. Material and methods.
    The study design should be included.
    In line 121 the authors should avoid mentioning table 3 before table 2. Tables should be mentioned in the text in order of appearance.

Author Response

Reviewer 3 comment: First, I would like to thank you for trusting me to review your manuscript. It is essential to determine the existing strategies for the promotion of physical activity carried out by institutions and to estimate whether these are adequate for that purpose or can be improved. In any case, there are some questions that should be considered prior to publication. 

Our response: Thank you for the positive review of our manuscript. We hope we have addressed all of your comments appropriately. 

Reviewer 3 comment: Introduction- Line 47: the first time the authors mention AT they should explain that it is an active transport.

Our response: Thank you, we have edited this in the first paragraph

Reviewer 3 comment: Line 87: why do the authors mention a study conducted in Sweden? The authors should justify the connection to the study area

Our response: Thank you for this comment, we have edited the text to clarify

Reviewer 3 comment: Material and methods

The study design should be included.

Our response: Thank you for this comment, additional details about the study design and methodology have been added to the Materials and Methods section (lines 132-140 and 150-151).

Reviewer 3 comment: In line 121 the authors should avoid mentioning table 3 before table 2. Tables should be mentioned in the text in order of appearance.

Our response: Thank you for noticing this, we have corrected this error.